# Pharmacologic activation of integrated stress response kinases inhibits pathologic mitochondrial fragmentation

Kelsey R Baron[1,†], Samantha Oviedo[1,2,†], Sophia Krasny[1], Mashiat Zaman[3], Rama Aldakhlallah[1], Prerona Bora[1], Prakhyat Mathur[1], Gerald Pfeffer[4,5], Michael J Bollong[6], Timothy E Shutt[7], Danielle A Grotjahn[2]*, R Luke Wiseman[1]*

[1]Department of Molecular and Cellular Biology, The Scripps Research Institute, La Jolla, United States; [2]Department of Integrative Structural and Computation Biology, The Scripps Research Institute, La Jolla, United States; [3]Department of Biochemistry and Molecular Biology, Cummings School of Medicine, University of Calgary, Calgary, Canada; [4]Hotchkiss Brain Institute, Department of Clinical Neurosciences, Cumming School of Medicine, University of Calgary, Calgary, Canada; [5]Alberta Child Health Research Institute, Department of Medical Genetics, Cumming School of Medicine, University of Calgary, Calgary, Canada; [6]Department of Chemistry, The Scripps Research Institute, La Jolla, United States; [7]Departments of Medical Genetics and Biochemistry & Molecular Biology, Cumming School of Medicine, Hotchkiss Brain Institute, Snyder Institute for Chronic Diseases, Alberta Children's Hospital Research Institute, University of Calgary, Calgary, Canada

*For correspondence:
grotjahn@scripps.edu (DAG);
wiseman@scripps.edu (RLW)

[†]These authors contributed equally to this work

## eLife Assessment

This **important** study identifies a new class of small molecules that activate the integrated stress response (ISR) via the kinase HRI. **Convincing** evidence, including the image analysis pipeline, indicates that two of these compounds promote mitochondrial elongation and protect against mitochondrial fragmentation caused by chemical stress conditions or by genetic alterations. These findings open an avenue for new strategies for mitochondrial dysfunction targeting linked to ISR alterations.

**Abstract** Excessive mitochondrial fragmentation is associated with the pathologic mitochondrial dysfunction implicated in the pathogenesis of etiologically diverse diseases, including many neurodegenerative disorders. The integrated stress response (ISR) – comprising the four eIF2α kinases PERK, GCN2, PKR, and HRI – is a prominent stress-responsive signaling pathway that regulates mitochondrial morphology and function in response to diverse types of pathologic insult. This suggests that pharmacologic activation of the ISR represents a potential strategy to mitigate pathologic mitochondrial fragmentation associated with human disease. Here, we show that pharmacologic activation of the ISR kinases HRI or GCN2 promotes adaptive mitochondrial elongation and prevents mitochondrial fragmentation induced by the calcium ionophore ionomycin. Further, we show that pharmacologic activation of the ISR reduces mitochondrial fragmentation and restores basal mitochondrial morphology in patient fibroblasts expressing the pathogenic D414V variant of the pro-fusion mitochondrial GTPase MFN2 associated with neurological dysfunctions, including ataxia, optic atrophy, and sensorineural hearing loss. These results identify pharmacologic activation of ISR kinases as a potential strategy to prevent pathologic mitochondrial fragmentation induced by disease-relevant chemical and genetic insults, further motivating the pursuit of highly selective ISR

kinase-activating compounds as a therapeutic strategy to mitigate mitochondrial dysfunction implicated in diverse human diseases.

## Introduction

The integrated stress response (ISR) comprises four stress-regulated kinases – PERK, PKR, GCN2, and HRI – that selectively phosphorylate eIF2α in response to diverse pathologic insults (*Costa-Mattioli and Walter, 2020*; *Pakos-Zebrucka et al., 2016*). The ISR has recently emerged as a prominent stress-responsive signaling pathway activated by different types of mitochondrial stress (*Anderson and Haynes, 2020*; *Quirós et al., 2016*; *Winter et al., 2022*). Mitochondrial uncoupling or inhibition of ATP synthase activates the ISR through a mechanism involving the cytosolic accumulation and oligomerization of the mitochondrial protein DELE1, which binds to and activates the ISR kinase HRI (*Fessler et al., 2020*; *Fessler et al., 2022*; *Guo et al., 2020*; *Sekine et al., 2023*; *Yang et al., 2023a*). Complex I inhibition can also activate the ISR downstream of the ISR kinase GCN2 (*Mick et al., 2020*). Further, CRISPRi-depletion of mitochondrial proteostasis factors preferentially activates the ISR over other stress-responsive signaling pathways (*Madrazo et al., 2024*). These results identify the ISR as an important stress-responsive signaling pathway activated in response to many different mitochondrial insults.

Consistent with its activation by mitochondrial stress, the ISR regulates many aspects of mitochondrial biology. ISR kinases are activated in response to stress through a conserved mechanism involving dimerization and autophosphorylation (*Costa-Mattioli and Walter, 2020*; *Pakos-Zebrucka et al., 2016*). Once activated, these kinases phosphorylate eIF2α, resulting in both a transient attenuation of new protein synthesis and the activation of stress-responsive transcription factors such as ATF4 (*Costa-Mattioli and Walter, 2020*; *Pakos-Zebrucka et al., 2016*). The ISR promotes adaptive remodeling of mitochondrial pathways through both transcriptional and translational signaling. For example, the activation of ATF4 downstream of phosphorylated eIF2α transcriptionally regulates the expression of mitochondrial chaperones (e.g., the mitochondrial HSP70 *HSPA9*) and proteases (e.g., the AAA+ protease *LONP1*), increasing mitochondrial proteostasis capacity during stress (*Hori et al., 2002*; *Han et al., 2013*). ISR-dependent translational attenuation also enhances mitochondrial proteostasis through the selective degradation of the core import subunit TIM17A, which slows mitochondrial protein import and reduces the load of newly synthesized proteins entering mitochondria during stress (*Rainbolt et al., 2013*). Apart from proteostasis, transcriptional and translational signaling induced by the activation of ISR kinases also regulates many other mitochondrial functions, including phospholipid synthesis, cristae organization, and electron transport chain activity (*Balsa et al., 2019*; *Barad et al., 2023*; *Latorre-Muro et al., 2021*; *Lebeau et al., 2018*; *Perea et al., 2023b*).

Intriguingly, the morphology of mitochondria is also regulated by ISR signaling. Mitochondrial morphology is dictated by the relative activities of pro-fission and pro-fusion GTPases localized to the outer and inner mitochondrial membranes (OMM and IMM, respectively) (*Chan, 2020*; *Quintana-Cabrera and Scorrano, 2023*). These include the pro-fusion GTPases MFN1 and MFN2 and the pro-fission GTPase DRP1, all localized to the OMM. Previous results showed that ER stress promotes adaptive mitochondrial elongation downstream of the PERK arm of the ISR through a mechanism involving the accumulation of the phospholipid phosphatidic acid on the OMM where it inhibits the pro-fission GTPase DRP1 (*Lebeau et al., 2018*; *Perea et al., 2023b*; *Perea et al., 2023a*). This PERK-dependent increase of mitochondrial elongation functions to protect mitochondria during ER stress through multiple mechanisms, including enhanced regulation of respiratory chain activity, suppression of mitochondrial fragmentation, and reductions in the turnover of mitochondria by mitophagy (*Lebeau et al., 2018*; *Perea et al., 2023b*; *Perea et al., 2023a*). Apart from PERK, pharmacologic activation of the alternative ISR kinase GCN2 also promotes adaptive mitochondrial elongation (*Perea et al., 2023a*). This indicates that mitochondrial elongation may be a protective mechanism that can be pharmacologically accessed by activating multiple different ISR kinases.

Excessive mitochondrial fragmentation is a pathologic hallmark of numerous human diseases, including several neurodegenerative disorders (*Chan, 2020*; *Sharma et al., 2021a*; *Chen et al., 2023*; *Liu et al., 2020*; *Sprenger and Langer, 2019*). Fragmented mitochondria are often associated with mitochondrial dysfunctions, including impaired respiratory chain activity and dysregulation of mitophagy (*Chan, 2020*; *Sharma et al., 2021a*; *Chen et al., 2023*; *Liu et al., 2020*; *Sprenger and Langer,*

*2019*; *Giacomello et al., 2020*; *Wai and Langer, 2016*). Thus, pathologic increases in mitochondrial fragmentation are linked to many mitochondrial dysfunctions implicated in human disease. Consistent with this observation, interventions that prevent mitochondrial fragmentation have been shown to correct pathologic mitochondrial dysfunction in models of many different diseases (*Whitley et al., 2019*; *Bhatti et al., 2023*). Notably, genetic or pharmacologic inhibition of the pro-fission GTPase DRP1 blocks pathologic mitochondrial fragmentation and subsequent mitochondrial dysfunction in cellular and in vivo models of diverse neurodegenerative diseases, including Charcot–Marie–Tooth (CMT) Type 2A and Type 2B, spastic paraplegia, optic atrophy, Huntington's disease, amyotrophic lateral sclerosis, and Alzheimer's disease (*Gu et al., 2022*; *Das et al., 2022*; *Chen et al., 2022*; *Yang et al., 2023b*; *Guo et al., 2013*; *Joshi et al., 2018*; *Bera et al., 2022*; *Manczak et al., 2016*; *Kandi-malla et al., 2022*; *Yan et al., 2015*; *Baek et al., 2017*; *Mou and Li, 2019*).

The ability of ISR activation to promote mitochondrial elongation suggests that pharmacologic activation of ISR kinases could mitigate the pathologic mitochondrial fragmentation implicated in etiologically diverse human diseases. To test this idea, we probed the potential for pharmacologic activation of different ISR kinases to reduce mitochondrial fragmentation induced by disease-relevant chemical and genetic insults. Previous work identified the small-molecule halofuginone as a potent activator of the ISR kinase GCN2 that promotes ISR-dependent, adaptive mitochondrial elongation (*Perea et al., 2023a*; *Keller et al., 2012*). However, few other compounds were available that selectively activated other ISR kinases and were suitable for probing mitochondrial adaptation induced by ISR kinase activation (*Perea et al., 2023a*). Here, we used a small-molecule screening platform to identify two nucleoside mimetic compounds, 0357 and 3610, that preferentially activate the ISR downstream of the ISR kinase HRI. Using these compounds, we demonstrate that pharmacologic HRI activators also promote adaptive, ISR-dependent mitochondrial elongation. We go on to show that pharmacologic activation of the ISR with these compounds prevents DRP1-mediated mitochondrial fragmentation induced by the calcium ionophore ionomycin (*Ji et al., 2015*; *Ji et al., 2017*). Further, we show that compound-dependent activation of the ISR reduces the population of fragmented mitochondria and rescues basal mitochondrial network morphology in patient fibroblasts expressing the D414V variant of the pro-fusion GTPase MFN2 associated with a complex clinical phenotype including ataxia, optic atrophy, and sensorineural hearing loss (*Sharma et al., 2021b*). These results demonstrate the potential for pharmacologic activation of the ISR to prevent pathologic mitochondrial fragmentation in disease-relevant models. Moreover, our work further motivates the continued development of highly selective activators of ISR kinases as a potential therapeutic strategy to mitigate the pathologic mitochondrial dysfunction implicated in etiologically diverse human diseases.

## Results

### The nucleoside mimetic compounds 0357 and 3610 preferentially activate the ISR downstream of HRI

The small-molecule halofuginone activates the ISR kinase GCN2 and induces ISR-dependent mitochondrial elongation (*Perea et al., 2023a*; *Keller et al., 2012*) However, few other compounds are available that selectively activate other ISR kinases through mechanisms that allow for ISR-dependent mitochondrial remodeling (*Perea et al., 2023a*). For example, BtdCPU activates the ISR kinase HRI through a mechanism involving mitochondrial uncoupling, precluding its use for probing ISR-dependent protection of mitochondria (*Perea et al., 2023a*). To address this limitation and define the potential for pharmacologic activation of other ISR kinases to promote adaptive mitochondrial elongation, we established and implemented a screening platform to identify compounds that activated ISR signaling downstream of alternative ISR kinases (*Figure 1A*). In this screen, we used the ATF4-FLuc translational reporter of the ISR (*Figure 1—figure supplement 1A*; *Yang et al., 2023a*) We confirmed that ISR-activating stressors including the ER stressor thapsigargin (Tg) and the ATP synthase inhibitor oligomycin A (OA) robustly activated this reporter (*Figure 1—figure supplement 1B*). We then used this reporter to screen the ~3k nucleoside mimetic analog compound library (10 μM) and monitored ATF4-FLuc activity 8 hr after treatment. Our primary screen identified 34 hit compounds that activated the ATF4-FLuc reporter with a robust Z-score of greater than 3 fold. We then removed highly reactive compounds and pan-assay interference compounds (PAINS), which reduced the number of hits to 9 (*Figure 1B*). These compounds were re-purchased and then

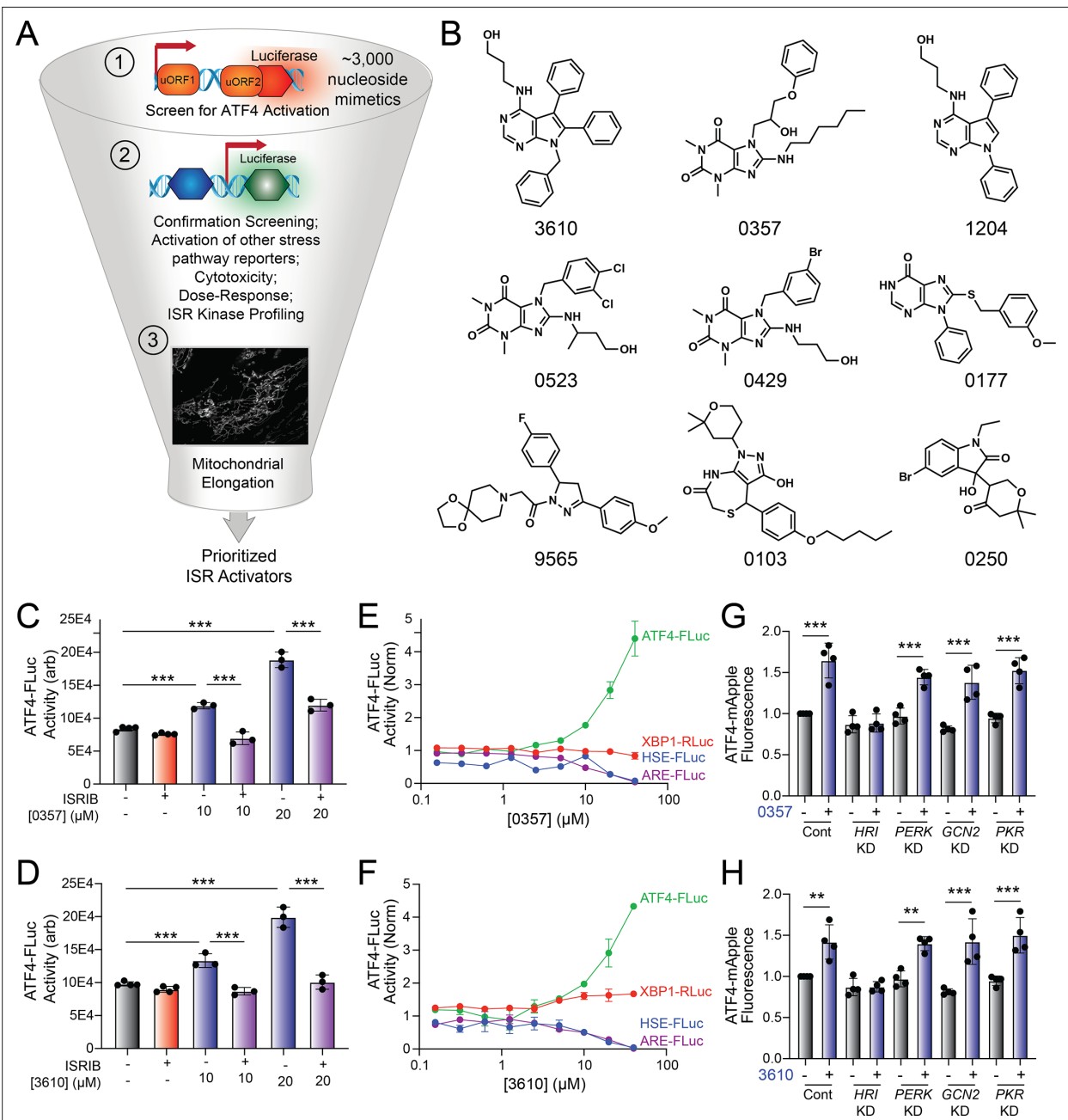

**Figure 1.** Identification of nucleoside mimetics that preferentially activate the integrated stress response (ISR) kinase HRI. (**A**) Screening pipeline used to identify selective ISR kinase-activating compounds that promote protective mitochondrial elongation. (**B**) Structures of the top 9 ISR-activating compounds identified in our nucleoside mimetic screen. (**C, D**) ATF4-FLuc activity in HEK293 cells stably expressing ATF4-FLuc (*Yang et al., 2023a*) treated for 8 hr with the indicated concentration of 0357 (**C**) or 3610 (**D**) in the absence or presence of ISRIB (200 nM). (**E, F**) Activation of the ATF4-FLuc ISR translational reporter (green), the XBP1-RLuc UPR reporter (red), the HSE-FLuc HSR reporter (blue), or the ARE-FLuc OSR reporter (purple) stably expressed in HEK293 cells treated with the indicated concentration of 0357 (**E**) or 3610 (**F**) for 16 hr. (**G, H**) ATF4-mAPPLE fluorescence in HEK293 cells stably expressing ATF4-mAPPLE and CRISPRi depleted of the indicated ISR kinase (*Guo et al., 2020*) treated for 8 hr with 0357 (**G**, 20 μM) or 3610 (**H**, 20 μM). **p<0.01, ***p<0.005 for one-way ANOVA.

The online version of this article includes the following source data and figure supplement(s) for figure 1:

**Source data 1.** Excel spreadsheet containing source data for panels C-H.

**Figure supplement 1.** Identification of nucleoside mimetics that preferentially activate the integrated stress response (ISR) kinase HRI.

**Figure supplement 1—source data 1.** Excel spreadsheet containing source data for panels C-F.

tested in dose response for ATF4-FLuc activation (*Figure 1—figure supplement 1C*). This identified compounds 0357 and 3610 as the compounds that most efficaciously activated the ATF4-FLuc reporter, albeit with low potency ($EC_{50} > 10$ μM). We confirmed that co-treatment with the highly selective ISR inhibitor ISRIB (*Sidrauski et al., 2013*; *Zyryanova et al., 2018*) blocked ATF4-FLuc activation induced by these compounds, confirming this activation can be attributed to the ISR (*Figure 1C and D*). Further, we used qPCR to show that treatment with 0357 or 3610 increased expression of the ISR target genes *ASNS* and *CHAC1* in HEK293 and MEF cells (*Figure 1—figure supplement 1D and E*; *Kreß et al., 2023*; *Grandjean et al., 2019*). Importantly, these compounds did not activate luciferase reporters of other stress-responsive signaling pathway such as the unfolded protein response (UPR; XBP1-RLuc) (*Grandjean et al., 2020*; *Plate et al., 2016*), the heat shock response (HSR; HSE-FLuc) (*Calamini et al., 2011*), or the oxidative stress response (OSR; ARE-FLuc) (*Ibrahim et al., 2020*; *Figure 1E and F*). Further, treatment with these compounds did not induce expression of the UPR target gene *BiP*, the HSR target gene *HSPA1A*, or the OSR target gene *NQO1* in HEK293 cells (*Figure 1—figure supplement 1F*). Finally, treatment with 0357 or 3610 did not significantly reduce viability of HEK293 cells (*Figure 1—figure supplement 1G*). These results indicate compounds 0357 and 3610 preferentially activate the ISR compared to other stress-responsive signaling pathways.

Next, we sought to identify the specific ISR kinase responsible for ISR activation induced by these two nucleoside mimetics. We monitored the compound-dependent activation of an ATF4-mAPPLE fluorescent reporter stably expressed in HEK293 cells CRISPRi-depleted of each individual ISR kinase (*Figure 1—figure supplement 1A*; *Guo et al., 2020*; *Perea et al., 2023a*). We previously used this assay to confirm that halofuginone activates the ISR downstream of GCN2 and BtdCPU activates the ISR downstream of HRI (*Perea et al., 2023a*). Treatment with either 0357 or 3610 activates the ATF4-mAPPLE reporter in control cells (*Figure 1G and H*). CRISPRi-depletion of *HRI*, but no other ISR kinase, blocked ATF4-FLuc activation induced by these two compounds. This finding indicates that these compounds activate the ISR downstream of HRI. Collectively, these results identify 0357 and 3610 as nucleoside mimetic compounds that preferentially activate the ISR through a mechanism involving the ISR kinase HRI.

## Pharmacologic HRI activators promote mitochondrial elongation

Halofuginone-dependent activation of GCN2 promotes adaptive mitochondrial elongation (*Perea et al., 2023a*). However, it is currently unclear if pharmacologic activation of other ISR kinases can similarly induce mitochondrial elongation. Here, we tested the ability of our HRI-activating compounds 0357 and 3610 to induce mitochondrial elongation downstream of the ISR. Previously, ISR-dependent mitochondrial elongation was quantified by manually classifying cells as containing fragmented, tubular, or elongated networks (*Lebeau et al., 2018*; *Perea et al., 2023b*). However, since 0357 and 3610 activate ISR signaling to lower levels than that observed for other compounds (e.g., halofuginone), we posited that these compounds may induce more modest mitochondrial elongation that may be difficult to quantify using this manual approach. To address this, we implemented an automated image analysis pipeline using Imaris software to quantify mitochondrial elongation in MEF cells stably expressing mitochondrial-targeted GFP (mtGFP) treated with our compounds (*Figure 2—figure supplement 1A*); (*Wang et al., 2012*). We collected Z-stack confocal images of mtGFP-expressing MEF cells (MEFmtGFP) and processed images using a deconvolution filter in FIJI to reduce the background and enhance the fluorescent signal. We used the 'Surfaces' module on Imaris to generate three-dimensional (3D) segmentation models of mitochondria visible in the deconvolved Z-stacks. Using the surfaces module, we quantified parameters defining mitochondrial shape, including bounding box length, sphericity, and ellipsoid principal axis length (*Figure 2—figure supplement 1A*). Treatment with conditions that induce mitochondrial elongation, such as the ER stressor thapsigargin (Tg) and the GCN2 activator halofuginone (HF), increased bounding box and ellipsoid principal axis length, while reducing sphericity (*Figure 2A–D*, *Figure 2—figure supplement 1B–D*) – all changes consistent with increases in mitochondrial length. In contrast, treatment with conditions that promote mitochondrial fragmentation, such as the mitochondrial uncouplers BtdCPU and CCCP (both compounds that activate HRI) (*Fessler et al., 2020*; *Perea et al., 2023a*), reduced bounding box length and ellipsoid principal axis length, while increasing sphericity (*Figure 2A–D*, *Figure 2—figure supplement 1B–D*) – all changes consistent with increased mitochondrial fragmentation.

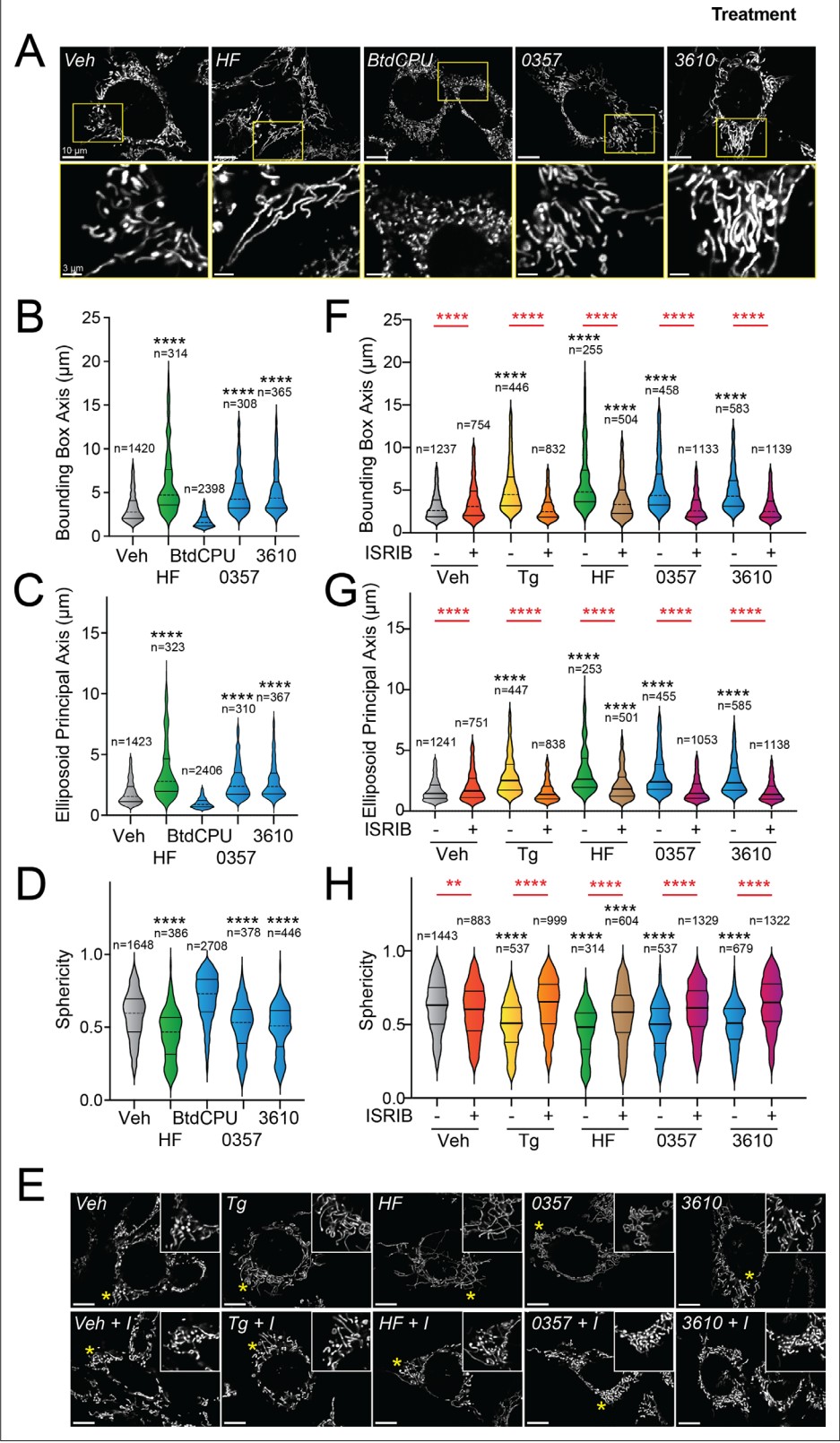

**Figure 2.** Pharmacologic HRI activation induces integrated stress response (ISR)-dependent mitochondrial elongation. (**A**) Representative images of MEF cells stably expressing ᵐᵗGFP (MEFᵐᵗGFP) (**Wang et al., 2012**) treated for 6 hr with vehicle (veh), halofuginone (HF, 100 nM), BtdCPU (10 µM), 0357 (10 µM), or 3610 (10 µM). The inset shows a 3-fold magnification of the region indicated by the yellow box. Scale bars, 10 µm (top) and 3.33 µm

*Figure 2 continued on next page*

*Figure 2 continued*

(bottom). (**B–D**) Quantification of bounding box axis, ellipsoid principal axis, and sphericity from the entire dataset of representative images shown in (**A**). The number of individual measurements for each condition is shown above. (**E**) Representative images of MEF^mtGFP cells treated for 6 hr with vehicle (veh), thapsigargin (Tg; 0.5 μM) halofuginone (HF, 100 nM), 0357 (10 μM), or 3610 (10 μM) in the presence or absence of ISRIB (200 nM). The inset shows 3-fold magnification of the image centered on the asterisks. Scale bars, 10 μm. (**F–H**) Quantification of bounding box axis, ellipsoid principal axis, and sphericity from the entire dataset of representative images shown in (**E**). The number of 3D segmentations used for the individual measurements for each condition is shown above. *p<0.05, ****p<0.001 for Kruskal–Wallis ANOVA. Black asterisks indicate comparison to vehicle-treated cells. Red asterisks show comparisons for ISRIB co-treatment.

The online version of this article includes the following source data and figure supplement(s) for figure 2:

**Source data 1.** Excel spreadsheet containing source data for panels B-D and F-H.

**Figure supplement 1.** Pharmacologic HRI activation induces integrated stress response (ISR)-dependent mitochondrial elongation.

**Figure supplement 1—source data 1.** Excel spreadsheet containing source data for panels C-E.

---

We next applied this approach to define the impact of our pharmacologic HRI activators on mitochondrial morphology. Treatment with either 0357 or 3610 for 6 hr increased both the bounding box and ellipsoid principal axis length, while decreasing mitochondrial sphericity (*Figure 2A–D*). This result indicates that both these compounds induced mitochondrial elongation. Co-treatment with the selective ISR inhibitor ISRIB blocked these changes in mitochondria shape, indicating that these compounds induce mitochondrial elongation through an ISR-dependent mechanism (*Figure 2E–H*). ISRIB co-treatment also blocked mitochondrial elongation induced by the ER stressor thapsigargin (Tg) and the GCN2 activator halofuginone (HF), as predicted (*Lebeau et al., 2018*; *Perea et al., 2023a*). These results indicate that, like halofuginone, pharmacologic HRI activators also induce adaptive, ISR-dependent mitochondrial elongation. Further, these results suggest that the pharmacologic activation of different ISR kinases can induce protective elongation of mitochondria in the absence of cellular stress.

## Pharmacologic ISR activation suppresses ionomycin-induced mitochondrial fragmentation

The ability for our pharmacologic GCN2 or HRI activators to induce adaptive mitochondrial elongation suggests that enhancing signaling through these kinases may suppress mitochondrial fragmentation induced by pathologic insults such as calcium dysregulation (*Calvo-Rodriguez and Bacskai, 2020*; *Calvo-Rodriguez et al., 2020*; *Garbincius and Elrod, 2022*; *Matuz-Mares et al., 2022*). Treatment with the calcium ionophore ionomycin induces rapid, DRP1-dependent mitochondrial fragmentation in cell culture models (*Ji et al., 2015*; *Ji et al., 2017*). We pretreated MEF^mtGFP cells for 6 hr with the GCN2 activator halofuginone or our two HRI-activating compounds (0357 and 3610) and subsequently challenged these cells with ionomycin. We then monitored mitochondrial morphology over a 15 min timecourse. As expected, ionomycin rapidly increased the accumulation of fragmented mitochondria in these cells, evidenced by reductions in both bounding box and ellipsoid principal axis length and increases of organelle sphericity (*Figure 3—figure supplement 1A–D*). Pretreatment with the ER stressor thapsigargin, which promotes stress-induced mitochondrial elongation downstream of the PERK ISR kinase, reduced the accumulation of fragmented mitochondria in ionomycin-treated cells (*Figure 3A–D*), as previously reported (*Perea et al., 2023b*). Intriguingly, treatment with halofuginone, 0357, or 3610 also reduced the accumulation of fragmented mitochondria in ionomycin-treated cells. Instead, mitochondria in cells pretreated with these ISR kinase activators and challenged with ionomycin demonstrated mitochondrial lengths and sphericity similar to that observed in vehicle-treated MEF^mtGFP cells (*Figure 3A–D*). These results indicate that pharmacologic activation of different ISR kinases can suppress the accumulation of fragmented mitochondria following ionomycin-induced calcium dysregulation.

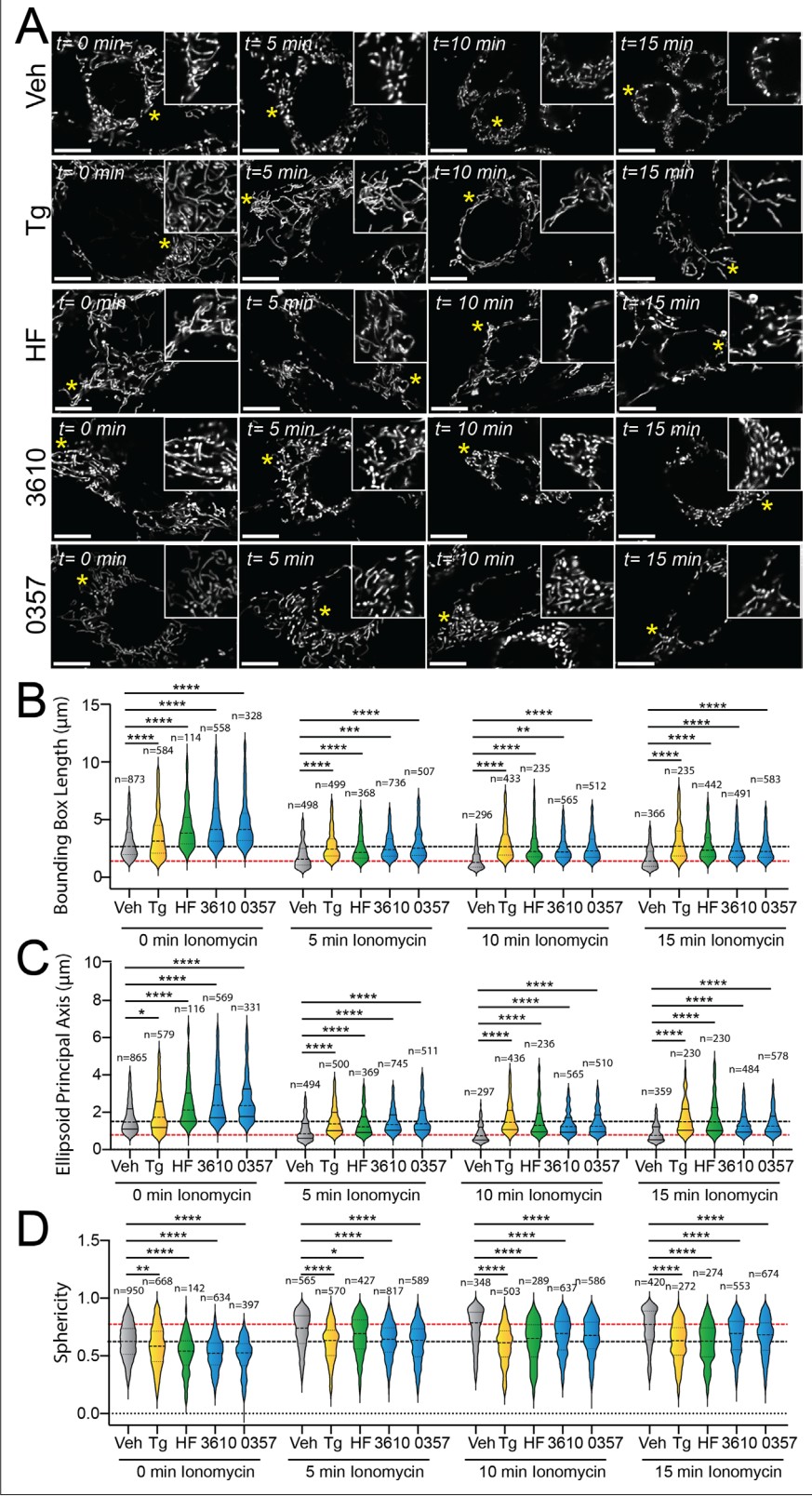

**Figure 3.** Pharmacologic activation of integrated stress response (ISR) kinases prevents ionomycin-dependent accumulation of fragmented mitochondria. (**A**) Representative images of MEF^mtGFP cells pretreated for 6 hr with vehicle (veh), thapsigargin (Tg, 500 nM), halofuginone (HF, 100 nM), 0357 (10 μM), or 3610 (10 μM) and then challenged with ionomycin (1 μM) for the indicated time. The inset shows 3-fold magnification of the image

*Figure 3 continued on next page*

*Figure 3 continued*

centered on the asterisk. Scale bars, 10 µm. (**B–D**) Quantification of bounding box axis length, ellipsoid principal axis length, and sphericity from the entire dataset of representative images shown in (**A**). The black dashed line shows the mean value of vehicle-treated cells prior to ionomycin treatment. The dashed red line shows the mean value of vehicle-treated cells following 15 min treatment with ionomycin. The number of 3D segmentations used for the individual measurements for each condition is shown above. *p<0.05, **p<0.01, ***p<0.005, ****p<0.001 for Kruskal–Wallis ANOVA. Black asterisks show comparison with vehicle-treated cells.

The online version of this article includes the following source data and figure supplement(s) for figure 3:

**Source data 1.** Excel spreadsheet containing source data for panels B-D.

**Figure supplement 1.** Pharmacologic activation of integrated stress response (ISR) kinases prevents ionomycin-dependent accumulation of fragmented mitochondria.

**Figure supplement 1—source data 1.** Excel spreadsheet containing source data for panels A-C.

## Pharmacologic activation of ISR kinases restores basal mitochondrial morphology in patient fibroblasts expressing disease-associated MFN2$^{D414V}$

Over 150 pathogenic variants in the pro-fusion GTPase MFN2 are causatively associated with the autosomal-dominant peripheral neuropathy CMT Type 2A (*Cartoni and Martinou, 2009*; *Zaman and Shutt, 2022*; *Alberti et al., 2024*). While these pathogenic variants can impact diverse aspects of mitochondrial biology (*Zaman and Shutt, 2022*), many, including D414V, lead to increases in mitochondrial fragmentation (*Sharma et al., 2021b*; *Cartoni and Martinou, 2009*; *Zaman and Shutt, 2022*; *Alberti et al., 2024*). This can be attributed to reduced activity of MFN2-dependent fusion associated with these variants and a subsequent relative increase of DRP1-dependent mitochondrial fission. We predicted that pharmacologic activation of ISR kinases could rescue mitochondrial network morphology in patient fibroblasts expressing the disease-associated MFN2 variant D414V (MFN2$^{D414V}$). To test this, we treated wild-type human fibroblasts and patient fibroblasts expressing MFN2$^{D414V}$ with halofuginone or our two HRI-activating compounds 0357 and 3610 and monitored mitochondrial network morphology by staining with MitoTracker. As reported previously, MFN2$^{D414V}$-expressing fibroblasts showed shorter, more fragmented mitochondrial networks compared to control fibroblasts, reflected by reductions in both bounding box and ellipsoid principle axis lengths and increased sphericity (*Figure 4A and E*, *Figure 4—figure supplement 1A–C*; *Sharma et al., 2021b*). Treatment with halofuginone, 0357, or 3610 increased mitochondrial length in control fibroblasts (*Figure 4A–D*). These changes were inhibited by co-treatment with ISRIB, confirming these effects can be attributed to ISR activation. Intriguingly, all three compounds also increased mitochondrial length and reduced sphericity in MFN2$^{D414V}$-expressing patient fibroblasts to levels similar to those observed in control fibroblasts, with halofuginone showing the largest effect (*Figure 4E–H*). Again, this increase in mitochondrial elongation was reversed by co-treatment with ISRIB. These results show that pharmacologic activation of different ISR kinases can rescue basal mitochondrial morphology in patient fibroblasts expressing the disease-associated MFN2$^{D414V}$ variant that causes dysregulation in various neurological functions, including ataxia, optic atrophy, and sensorineural hearing loss.

## Discussion

Here, we show that pharmacologic activation of different ISR kinases can prevent mitochondrial fragmentation induced by disease-relevant chemical or genetic insults. We identify two nucleoside mimetic compounds that preferentially activate the ISR downstream of the ISR kinase HRI. We demonstrate that these two HRI activators promote adaptive, ISR-dependent mitochondrial elongation. Further, we show that treatment with these HRI-activating compounds or the GCN2 activator halofuginone prevents the accumulation of fragmented mitochondria induced by Ca$^{2+}$ dysregulation and rescues mitochondrial network morphology in patient fibroblasts expressing the disease-associated, pathogenic D414V variant of MFN2. Collectively, these results demonstrate that pharmacologic activation of different ISR kinases can mitigate pathologic mitochondrial fragmentation induced by diverse disease-associated pathologic insults.

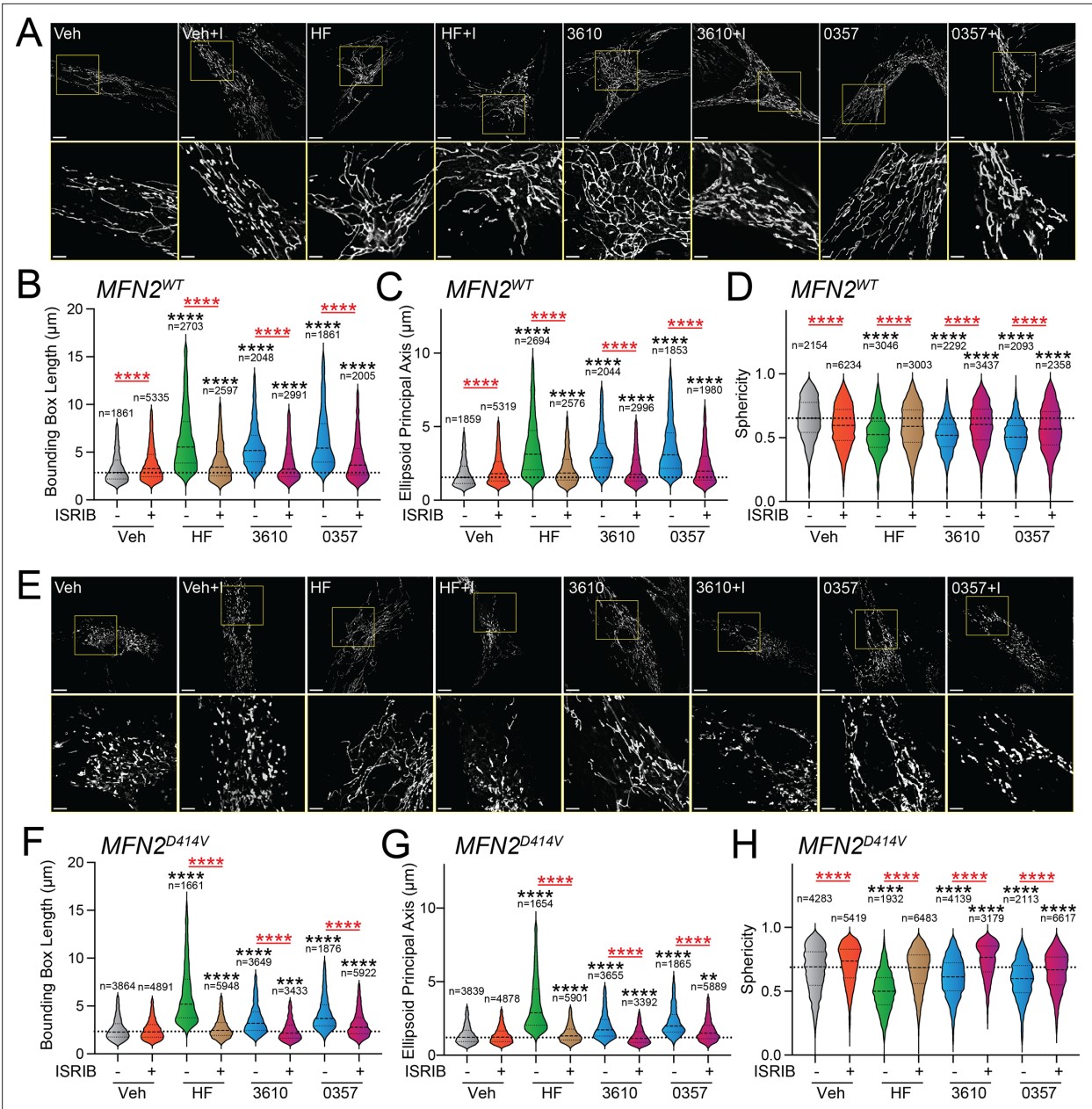

**Figure 4.** Pharmacologic activation of integrated stress response (ISR) kinases rescues basal mitochondrial morphology in patient fibroblasts expressing the disease-associated D414V MFN2 variant. (**A**) Representative images of control human fibroblasts expressing MFN2[WT] treated for 6 hr with vehicle (veh), halofuginone (HF, 100 nM), 3610 (10 μM), 0357 (10 μM), and/or ISRIB (200 nM). The inset shows 3-fold magnification of the image centered on the asterisks. Scale bars, 15 μm (top) and 5 μM (bottom). (**B–D**) Quantification of bounding box axis (**B**), ellipsoid principal axis (**C**), and sphericity (**D**) from the entire dataset of images described in panel (**A**). The number of individual measurements for each condition is shown above. (**E**) Representative images of patient fibroblasts expressing MFN2[D414V] treated for 6 hr with veh, HF (100 nM), 3610 (10 μM), 0357 (10 μM), and/or ISRIB (200 nM). The inset shows 3-fold magnification of the image centered on the asterisks. Scale bars, 15 μm (top) and 5 μM (bottom). (**F–H**) Quantification of bounding box axis (**F**), ellipsoid principal axis (**G**), and sphericity (**H**) from the entire dataset of images described in panel (**E**). The number of 3D segmentations used for the individual measurements for each condition is shown above. *p<0.05, ***p<0.005, ****p<0.001 for Kruskal–Wallis ANOVA. Black asterisks show comparison with vehicle-treated cells. Red asterisks show comparisons for ISRIB co-treatment.

The online version of this article includes the following source data and figure supplement(s) for figure 4:

**Source data 1.** Excel spreadsheet containing source data for panels B-D and F-H.

**Figure supplement 1.** Pharmacologic activation of integrated stress response (ISR) kinases rescue basal mitochondrial morphology in patient fibroblasts expressing the disease-associated D414V MFN2 variant.

**Figure supplement 1—source data 1.** Excel spreadsheet containing source data for panels A-C.

Mitochondrial fragmentation is often induced through a mechanism involving stress-dependent increases in the activity of the pro-fission GTPase DRP1 (*Chan, 2020*; *Giacomello et al., 2020*). This has motivated efforts to identify pharmacologic inhibitors of DRP1 to prevent mitochondrial fragmentation and subsequent organelle dysfunction associated with human disease (*Bhatti et al., 2023*; *Bera et al., 2022*; *Yu et al., 2023*). Activation of the ISR kinase PERK during ER stress promotes mitochondrial elongation through a mechanism involving inhibition of DRP1 (*Perea et al., 2023b*). This suggested that pharmacologic activation of other ISR kinases could also potentially inhibit DRP1 and block pathologic mitochondrial dysfunction induced by increased DRP1 activity. Consistent with this idea, we show that pharmacologic activation of the ISR using both GCN2 and HRI-activating compounds blocks the accumulation of fragmented mitochondria in ionomycin-treated cells – a condition that promotes mitochondrial fragmentation through increased DRP1-dependent fission (*Ji et al., 2015*; *Ji et al., 2017*). These results support a model whereby pharmacologic activation of different ISR kinases promotes organelle elongation by inhibiting DRP1 and suggests that pharmacologic activation of ISR kinases can be broadly applied to quell mitochondrial fragmentation in the myriad diseases associated with overactive DRP1.

Pathogenic variants in mitochondrial-targeted proteins are causatively associated with the onset and pathogenesis of numerous neurodegenerative disorders, including CMT, spinocerebellar ataxia (SCA), and spastic paraplegia (*Chan, 2020*; *Sharma et al., 2021a*; *Gorman et al., 2016*; *Lightowlers et al., 2015*; *Song et al., 2021*; *Deshwal et al., 2020*). In many of these diseases, mitochondrial fragmentation is linked to pathologic mitochondrial dysfunctions, such as reduced respiratory chain activity, decreased mtDNA, and increased apoptotic signaling. Intriguingly, pharmacologic or genetic inhibition of mitochondrial fragmentation mitigates cellular and mitochondrial pathologies associated with many of these diseases, highlighting the critical link between mitochondrial fragmentation and disease pathogenesis (*Das et al., 2022*; *Chen et al., 2022*; *Yang et al., 2023b*; *Mou and Li, 2019*). Herein, we show that pharmacologic activation of the ISR mitigates mitochondrial fragmentation in patient fibroblasts homozygous for the disease-associated D414V MFN2 variant. This finding highlights the potential for pharmacologic ISR activation to restore basal mitochondrial network morphology in cells expressing pathogenic MFN2 variants, although further study is necessary to determine the impact of ISR activation in cells from heterozygous patients expressing pathogenic MFN2 variants associated with CMT2A, the most common clinical phenotype associated with this disease. Regardless, our results are of substantial interest because over 70% of cases of axonal CMT are associated with pathogenic MFN2 variants, and to date, no disease-modifying therapies are available for any genetic subtype of CMT (*Yoshimura et al., 2019*; *Pisciotta et al., 2021*). Apart from MFN2 variants, our results also highlight the potential for pharmacologic ISR kinase activation to broadly mitigate pathologic mitochondrial fragmentation associated with the expression of disease-related, pathogenic variants of other mitochondrial proteins, which we are continuing to explore.

Apart from morphology, ISR signaling also promotes adaptive remodeling of many other aspects of mitochondrial biology, including proteostasis, electron transport chain activity, phospholipid synthesis, and apoptotic signaling (*Hori et al., 2002*; *Han et al., 2013*; *Rainbolt et al., 2013*; *Balsa et al., 2019*; *Barad et al., 2023*; *Latorre-Muro et al., 2021*; *Lebeau et al., 2018*; *Perea et al., 2023b*). While we specifically focus on defining the potential for pharmacologic activation of ISR kinases to rescue mitochondrial morphology in disease models, the potential of ISR activation to promote adaptive remodeling of other mitochondrial functions suggests that pharmacologic activators of ISR kinases could more broadly influence cellular and mitochondrial biology in disease states through the adaptive remodeling of these other pathways. For example, we previously showed that halofuginone-dependent GCN2 activation restored cellular ER stress sensitivity and mitochondrial electron transport chain activity in cells deficient in the alternative ISR kinase PERK (*Perea et al., 2023a*) – a model of neurodegenerative diseases associated with reduced PERK activity such as PSP and AD (*Park et al., 2023*; *Yuan et al., 2018*). Compounds that activate other ISR kinases are similarly predicted to promote the direct remodeling of mitochondrial pathways to influence these and other functions. Thus, as we, and others, continue defining the impact of pharmacologic ISR kinase activation on mitochondrial function, we predict to continue revealing new ways in which activation of ISR kinases directly regulates many other aspects of mitochondrial function disrupted in human disease.

Several different compounds have previously been reported to activate specific ISR kinases (*Perea et al., 2023a*; *Ganz et al., 2020*; *Stockwell et al., 2012*; *Chen et al., 2011*; *Szaruga et al., 2023*;

*Carlson et al., 2023*; *Thomson et al., 2024*). However, the potential for many of these compounds to mitigate mitochondrial dysfunction in human disease is limited by factors including lack of selectivity for the ISR or a specific ISR kinase, off-target activities, or low therapeutic windows (*Perea et al., 2023a*). For example, the HRI activator BtdCPU activates the ISR through a mechanism involving mitochondrial uncoupling and subsequent mitochondrial fragmentation (*Perea et al., 2023a*). Here, we identify two nucleoside mimetic compounds that activate the ISR downstream of HRI and show selectivity for the ISR relative to other stress-responsive signaling pathways, providing new tools to probe mitochondrial remodeling induced by ISR kinase activation. However, the low potency of these compounds limits their translational potential to mitigate mitochondrial dysfunction in disease, necessitating the identification of new, highly selective ISR kinase-activating compounds. As we and others continue identifying next-generation ISR kinase activators with improved translational potential, it will be important to optimize the pharmacokinetics (PK) and pharmacodynamic (PD) profiles of these compounds to selectively enhance adaptive, protective ISR signaling in disease-relevant tissues, independent of maladaptive ISR signaling often associated with chronic, stress-dependent activation of this pathway (*Costa-Mattioli and Walter, 2020*; *Pakos-Zebrucka et al., 2016*). As described previously for activators the IRE1 arm of the UPR, such improvements can be achieved by defining optimized compound properties and dosing regimens to control the timing and extent of pathway activity (*Madhavan et al., 2022*). Thus, as we and others continue pursuing pharmacologic ISR kinase activation as a strategy to target mitochondrial dysfunction in disease, we anticipate that we will continue to learn more about the central role for this pathway in adapting mitochondria during stress and establish pharmacologic ISR kinase activation as a viable approach to treat mitochondrial dysfunction associated with etiologically diverse diseases.

# Materials and methods

## Key resources table

| Reagent type (species) or resource | Designation | Source or reference | Identifiers | Additional information |
|---|---|---|---|---|
| Cell line (human) | HEK293 | ATCC | | |
| Cell line (human) | HEK293 cells expressing XBP1-RLuc | Wiseman Lab (TSRI) | | *Grandjean et al., 2020*; *Plate et al., 2016* |
| Cell line (human) | HEK293 cells expressing ATF4-FLuc | Martin Kampmann's lab (UCSF) | | *Yang et al., 2023a* |
| Cell line (human) | HEK293 cells expressing ATF4-mAPPLE and CRISPRi-depleted of *PERK*, *GCN2*, *HRI*, or *PKR* | Martin Kampmann's lab (UCSF) | | *Guo et al., 2020* |
| Cell line (mouse) | MEF cells stably expressing mitochondrial-targeted GFP (MEF^mtGFP) | Peter Schultz's lab (TSRI) | | *Wang et al., 2012* |
| Cell line (human) | Primary fibroblasts from patients expressing WT or D414V MFN2 | University of Calgary | | *Sharma et al., 2021a*; *Martens et al., 2020* |
| Commercial assay or kit | Promega Bright-Glo substrate | Promega | | |
| Commercial assay or kit | Quick-RNA MiniPrepKit | Zymo Research | | |
| Commercial assay or kit | High-Capacity Reverse Transcription Kit | Applied Biosystems | | |
| Commercial assay or kit | Power SYBR Green PCR Master Mix | Applied Biosystems | | |
| Chemical compound, drug | Thapsigargin (Tg) | Fisher Scientific | 50-464-294 | |
| Chemical compound, drug | ISRIB | Sigma | SML0843 | |
| Chemical compound, drug | CCCP | Sigma | C2759 | |
| Chemical compound, drug | BtdCPU | Fisher | 32-489-210MG | |
| Chemical compound, drug | Halofuginone (HF) | Sigma | 50-576-3001 | |
| Chemical compound, drug | Oligomycin A | Selleck | S1478 | |
| Chemical compound, drug | MitoTracker Green | Life Technologies | M7514 | |

*Continued on next page*

*Continued*

| Reagent type (species) or resource | Designation | Source or reference | Identifiers | Additional information |
|---|---|---|---|---|
| Software, algorithm | Imaris 10.0 | Oxford Instruments | | 3-D Surface Rendering Module |

## Mammalian cell culture

HEK293 cells (purchased from ATCC), HEK293 cells stably expressing XBP1-RLuc (*Grandjean et al., 2020*; *Plate et al., 2016*), HEK293 cells stably expressing HSE-FLuc, HEK293 cells stably expressing ATF4-FLuc (*Yang et al., 2023a*), HEK293 cells stably expressing ATF4-mAPPLE (a kind gift from Martin Kampmann's lab) (*Yang et al., 2023a*) and CRISPRi-depleted of individual ISR kinases (*HRI*, *PKR*, *PERK*, *GCN2*; a kind gift from Martin Kampmann's lab at UCSF) (*Guo et al., 2020*), and MEF$^{mtGFP}$ (a kind gift from Peter Schultz) (*Wang et al., 2012*) were all cultured at 37°C and 5% $CO_2$ in DMEM (Corning-Cellgro) supplemented with 10% fetal bovine serum (FBS, Gibco), 2 mM L-glutamine (Gibco), 100 U/mL penicillin, and 100 mg/mL streptomycin (Gibco).

Primary fibroblast cells were isolated from partial thickness skin biopsy, as previously described (*Sharma et al., 2021b*; *Martens et al., 2020*), from a patient who provided written informed consent for research studies using human tissues (University of Calgary Conjoint Research Ethics Board REB17-0850). Cells were cultured in Medium Essential Media (11095080, Gibco), supplemented with 10% FBS (12483020, Gibco). Cells were maintained at 37°C and 5% $CO_2$. Clinical information regarding this participant was previously reported and included ataxia, optic atrophy, and sensorineural hearing loss (*Sharma et al., 2021b*). Exome sequencing in this participant identified a homozygous c.1241A>T variant in *MFN2* (predicted to cause p.(Asp414Val)) and no other pathogenic variants.

## Compounds and reagents

The compounds used in this study were purchased from the following sources: thapsigargin (Tg; Cat# 50-464-294 Fisher Scientific), ISRIB (Cat# SML0843, Sigma), CCCP (Cat# C2759, Sigma), BtdCPU (Cat# 32-489-210MG, Fisher), halofuginone (Cat#50-576-3001, Sigma), and oligomycin A (S1478, Selleck). The nucleoside mimetic library was purchased from Chem Div. Hit compounds were repurchased from Chem Div.

## Measurements of ISR activation in ATF4-reporter cell lines

HEK293 cells stably expressing the ATF4-FLuc, HSE-Fluc, or the ARE-Fluc reporter were seeded at a density of 15,000 cells per well in 384-well white plates with clear bottoms (Greiner). The following day, cells were treated with the indicated compound in triplicate at the indicated concentration for 8 hr. After treatment, an equal volume of Promega Bright-Glo substrate (Promega) was added to the wells and allowed to incubate at room temperature for 10 min. Luminescence was then measured using an Infinite F200 PRO plate reader (Tecan) with an integration time of 1000 ms. This assay was used to both screen the nucleoside mimetic library in triplicate and monitor the activity of hit compounds. HEK293 cells expressing the XBP1-RLuc reporter were tested using an analogous approach to that described above, monitoring RLuc activity using Renilla-Glo reagent (Promega), as previously described (*Grandjean et al., 2020*).

HEK293 cells stably expressing the ATF4-mApple reporter and CRISPRi-depleted of specific ISR kinases were seeded at a density of 300,000 cells per well in six-well TC-treated flat-bottom plates (Genesee Scientific). The cells were treated the next day for 16 hr with the compound at the indicated concentration. Cells were then washed twice with phosphate-buffered saline (PBS) and dissociated using TrypLE Express (Thermo Fisher). The cells were then resuspended in PBS and 5% FBS to neutralize the enzymatic reaction. Flow cytometry was performed on a Bio-Rad ZE5 Cell Analyzer monitoring mAPPLE fluorescence (568/592 nm) using the 561 nm green-yellow laser in combination with the 577/15 filter. Analysis was performed using FlowJo Software (BD Biosciences).

## Quantitative PCR

The relative mRNA expression of target genes was measured using quantitative RT-PCR. Cells were treated as indicated and then washed with PBS (Gibco). RNA was extracted using Quick-RNA Mini-PrepKit (Zymo Research) according to the manufacturer's protocol. RNA (500 ng) was then converted

to cDNA using the High-Capacity Reverse Transcription Kit (Applied Biosystems). qPCR reactions were prepared using Power SYBR Green PCR Master Mix (Applied Biosystems), and primers (below) were obtained from Integrated DNA Technologies. Amplification reactions were run in an ABI 7900HT Fast Real Time PCR machine with an initial melting period of 95°C for 5 min and then 45 cycles of 10 s at 95°C, 30 s at 60°C.

## qPCR primers

|  | Forward | Reverse |
| --- | --- | --- |
| *M. musculus Chac1* | TGACCCTCCTTGAAGACCGTGA | AGTGTCATAGCCACCAAGCACG |
| *M. musculus Rplp2* | TGTCATCGCTCAGGGTGTTG | AAGCCAAATCCCATGTCGTC |
| *M. musculus Asns* | CCAAGTTCAGTATCCTCTCC | TAATTTGCCACCTTTCTAGC |
| *H. sapiens RPLP0* | CCACGCTGCTGAACATGC | TCGAACACCTGCTGGATGAC |
| *H. sapiens ASNS* | ATCACTGTCGGGATGTACCC | TGATAAAAGGCAGCCAATCC |
| *H. sapiens CHAC1* | GTGGTGACGCTCCTTGAAGA | TTCAGGGCCTTGCTTACCTG |
| *H. sapiens HSPA1A* | GCTGATGATGGGGGTTAACA | GGAGGCGGAGTACA |
| *H. sapiens NQO1* | GCCTCCTTCATGGCATAGTT | GGACTGCACCAGAGCCAT |

## Fluorescence microscopy

MEF^mtGFP were seeded at a density of 15,000 cells/well in eight-chamber slides (Ibidi) coated with poly-D-lysine (Sigma) (*Wang et al., 2012*). The next day cells were treated with the indicated dose of compound for the indicated time. After treatment, cells were imaged on a Zeiss LSM 880 Confocal Laser Scanning Microscope equipped with a full incubation chamber for regulating temperature and $CO_2$ during live-cell imaging.

Patient fibroblasts were seeded at 70,000 cells per dish in 35 mm dishes with 20 mm glass bottoms (D35-20-1.5-N, Cellvis) for live-cell imaging. After 24 hr, the compound treatments were administered at the dosage and for the time points indicated in the figure legends. The mitochondrial network in patient fibroblasts was visualized using 100 nM MitoTracker Green (M7514, Life Technologies) (*Kumar et al., 2001*) for 45 min, following washing three times with culture media, according to the manufacturer's instructions. The Z-stack images were acquired of patient fibroblasts using an Olympus Spinning Disc Confocal System (Olympus SD OSR) equipped with the Olympus UPlanApo 60XTIRF/1.50 Oil Objective using the CellSense Dimensions software. Acquired Z-stacks were analyzed using AI Machine Learning Segmentation (Imaris), as detailed below.

## Quantification of mitochondrial morphology

The Z-stack confocal images were processed in FIJI to reduce the background noise and enhance the fluorescent signal. The processed images are then introduced into the developed quantification pipeline in Imaris imaging software. In this approach, mitochondria are segmented in 3D using the 'Surfaces' module with a machine-learning algorithm that has been iteratively trained to detect foreground and background pixels in each Z-stack, filling blank holes within segments to generate 3D surfaces. The generated surfaces are filtered to include those above a minimum threshold of 250 voxels. While direct length measurements cannot be obtained through the Imaris surface module, indirect measurements of mitochondrial length are inferred from three separate calculations, including (1) object-oriented bounding box axis, (2) ellipsoid axis length, and (3) object sphericity (see *Figure 2—figure supplement 1A*). The object-oriented bounding box axis is calculated by measuring the length of the longest or principal bounding-box length of the smallest object-oriented rectangular box that encloses each 3D segmentation. The ellipsoid axis length is calculated by measuring the length of the longest or principal axis of each 3D segmentation. Sphericity is calculated by dividing the longest axis of each 3D segmentation by the length of the perpendicular axis.

## Statistical methods

Data are presented as mean ± SEM or as violin plots showing the mean and quartiles for the indicated number of measurements. Outliers were removed from datasets describing bounding box length and principal axis length, as appropriate, using the ROUT outlier test in PRISM 10 (GraphPad, San Diego, CA). Normality of datasets from our imaging studies was tested in PRISM 10 (GraphPad) using D'Agostino & Pearson, Anderson–Darling, Shapiro–Wilk, and Kolmogorov–Smirnov tests. Statistics were calculated in PRISM 10 (GraphPad) and analyzed by one-way ANOVA with Tukey's multiple correction test, Kruskal–Wallis or Mann–Whitney tests for data exhibiting a non-normal distribution, as indicated in the accompanying figure legends. Indications of nonsignificant interactions were generally omitted for clarity.

## Materials availability statement

All materials detailed in this article can be provided by the authors upon reasonable request or purchased from the indicated supplier.

## Acknowledgements

We thank Jie Sun and Sergei Kutseikin for experimental support and Evan Powers for critical reading of the manuscript. We thank Kathy Spencer and Scott Henderson in the TSRI Microscopy Facility for their support on the confocal imaging and analysis described in this project. We would also like to thank Martin Kampmann (UCSF), Xiaoyan Guo (UConn), and Jonathan Lin (Stanford) for experimental resources and advice related to this project. This work was supported by the National Institutes of Health (NIH; NS095892, NS125674 to RLW), the Canadian Institutes of Health Research (TES), an NIH F30 Predoctoral Fellowship (AG081061 to KB), National Science Foundation Predoctoral Fellowships (to SO and RA), and the Hotchkiss Brain Institute International Recruitment Scholarship (MZ).

## Additional information

### Competing interests

R Luke Wiseman: Reviewing editor, eLife. The other authors declare that no competing interests exist.

### Funding

| Funder | Grant reference number | Author |
|---|---|---|
| National Institutes of Health | NS095892 | R Luke Wiseman |
| National Institutes of Health | NS125674 | Danielle A Grotjahn R Luke Wiseman |
| National Institutes of Health | AG081061 | Kelsey R Baron |
| National Science Foundation Graduate Research Fellowship Program | | Samantha Oviedo Rama Aldakhlallah |
| Hotchkiss Brain Institute, University of Calgary | | Mashiat Zaman |
| Canadian Institutes of Health Research | | Timothy E Shutt |

The funders had no role in study design, data collection and interpretation, or the decision to submit the work for publication.

### Author contributions

Kelsey R Baron, Samantha Oviedo, Conceptualization, Formal analysis, Investigation, Writing – original draft, Writing – review and editing; Sophia Krasny, Rama Aldakhlallah, Prerona Bora, Prakhyat

Mathur, Formal analysis, Investigation, Writing – review and editing; Mashiat Zaman, Conceptualization, Formal analysis, Investigation, Writing – review and editing; Gerald Pfeffer, Resources; Michael J Bollong, Formal analysis, Writing – review and editing; Timothy E Shutt, Danielle A Grotjahn, R Luke Wiseman, Conceptualization, Formal analysis, Supervision, Funding acquisition, Writing – original draft, Project administration, Writing – review and editing

**Author ORCIDs**
Prerona Bora  https://orcid.org/0000-0002-3489-6146
Prakhyat Mathur  https://orcid.org/0000-0002-3129-4465
Michael J Bollong  https://orcid.org/0000-0001-9439-1476
Timothy E Shutt  https://orcid.org/0000-0001-5299-2943
Danielle A Grotjahn  https://orcid.org/0000-0001-5908-7882
R Luke Wiseman  https://orcid.org/0000-0001-9287-6840

Reviewer #1 (Public review): https://doi.org/10.7554/eLife.100541.3.sa1
Reviewer #2 (Public review): https://doi.org/10.7554/eLife.100541.3.sa2
Reviewer #3 (Public review): https://doi.org/10.7554/eLife.100541.3.sa3
Author response https://doi.org/10.7554/eLife.100541.3.sa4

---

## Additional files

### Supplementary files
MDAR checklist

### Data availability
Source data files for Figure 1, Figure 2, Figure 3, Figure 4, Figure 1-figure supplement 1, Figure 2-figure supplement 1, Figure 3-figure supplement 1, and Figure 4 - figure supplement 1 contain the numerical data used to generate the corresponding figures.

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
