## [Editor Report · eLife Assessment]

This **important** study identifies a new class of small molecules that activate the integrated stress response (ISR) via the kinase HRI. **Convincing** evidence, including the image analysis pipeline, indicates that two of these compounds promote mitochondrial elongation and protect against mitochondrial fragmentation caused by chemical stress conditions or by genetic alterations. These findings open an avenue for new strategies for mitochondrial dysfunction targeting linked to ISR alterations.

---

## [Referee Report · Reviewer #1 (Public review)]

Summary:

This manuscript (Baron, Oviedo et al., 2024) builds on a previous study from the Wiseman lab (Perea, Baron et al., 2023) and describes the identification of novel nucleoside mimetics that activate the HRI branch of the ISR and drive mitochondrial elongation. The authors develop an image processing and analysis pipeline to quantify the effects of these compounds on mitochondrial networks and show that these HRI activators mitigate ionomycin driven mitochondrial fragmentation. They then show that these compounds rescue mitochondrial morphology defects in patient-derived MFN2 mutant cell lines.

Strengths:

The identification of new ISR modulators opens new avenues for biological discovery surrounding the interplay between mitochondrial form/function and the ISR, a topic that is of broad interest. Conceptually, this work suggests that such compounds might represent new potential therapeutics for certain mitochondrial disorders. Additionally, the development of a quantitative image analysis pipeline is valuable and has the potential to extract subtle effects of various treatments on mitochondrial morphology.

Weaknesses:

While the ISR modulators described here correct the morphology of mitochondria in MFN2.D414V mutant cells, the impact of these compounds on the function of mitochondria in the mutant cells remains unaddressed. Sharma et al., 2022 provide data for a deficit in mitochondrial OCR in MFN2.D414V cells which, if rescued by these compounds, would strengthen the argument that pharmacological ISR kinase activation is a strategy for targeting the functional consequences of the dysregulation of mitochondrial form.

---

## [Referee Report · Reviewer #2 (Public review)]

Summary.

Mitochondrial dysfunction is associated with a wide spectrum of genetic and age-related diseases. Healthy mitochondria form a dynamic reticular network and constantly fuse, divide, and move. In contrast, dysfunctional mitochondria have altered dynamic properties resulting in fragmentation of the network and more static mitochondria. It has recently been reported that different types of mitochondrial stress or dysfunction activate kinases that control the integrated stress response, including HRI, PERK and GCN2. Kinase activity results in decreased global translation and increased transcription of stress response genes via ATF4, including genes that encode mitochondrial protein chaperones and proteases (HSP70 and LON). In addition, the ISR kinases regulate other mitochondrial functions including mitochondrial morphology, phospholipid composition, inner membrane organization, and respiratory chain activity. Increased mitochondrial connectivity may be a protective mechanism that could be initiated by pharmacological activation of ISR kinases, as was recently demonstrated for GCN2.

A small molecule screening platform was used to identify nucleoside mimetic compounds that activate HRI. These compounds promote mitochondrial elongation and protect against acute mitochondrial fragmentation induced by a calcium ionophore. Mitochondrial connectivity is also increased in patient cells with a dominant mutation in MFN2 by treatment with the compounds.

Strengths:

(1) The screen leverages a well-characterized reporter of the ISR: translation of ATF4-FLuc is activated in response to ER stress or mitochondrial stress. Nucleoside mimetic compounds were screened for activation of the reporter, which resulted in the identification of nine hits. The two most efficacious in dose response tests were chosen for further analysis (0357 and 3610). The authors clearly state that the compounds have low potency. These compounds were specific to the ISR and did not activate the unfolded protein response or the heat shock response. Kinases activated in the ISR were systematically depleted by CRISPRi revealing that the compounds activate HRI.

(2) The status of the mitochondrial network was assessed with an Imaris analysis pipeline and attributes such as length, sphericity, and ellipsoid principal axis length were quantified. The characteristics of the mitochondrial network in cells treated with the compounds were consistent with increased connectivity. Rigorous controls were included. These changes were attenuated with pharmacological inhibition of the ISR.

(3) Treatment of cells with the calcium ionophore results in rapid mitochondrial fragmentation. This was diminished by pre-treatment with 0357 or 3610 and control treatment with thapsigargin and halofuginone.

(4) Pathogenic mutations in MFN2 result in the neurodegenerative disease Charcot-Marie-Tooth Syndrome Type 2A (CMT2A). Patient cells that express Mfn2-D414V possess fragmented mitochondrial networks and treatment with 0357 or 3610 increased mitochondrial connectivity in these cells.

Weaknesses:

The weakness is the limited analysis of cellular changes following treatment with the compounds.

(1) Unclear how 0357 or 3610 alter other aspects of cellular physiology. While this would be satisfying to know, it may be that the authors determined that broad, unbiased experiments such as RNAseq or proteomic analysis are not justified due to the limited translational potential of these specific compounds.

(2) There are many changes in Mfn2-D414V patient cells including reduced respiratory capacity, reduced mtDNA copy number, and fewer mitochondrial-ER contact sites. These experiments are relatively narrow in scope and quantifying more than mitochondrial structure would reveal if the compounds improve mitochondrial function, as is predicted by their model.

Comments on revisions:

Many reviewer concerns have been addressed or will be addressed in forthcoming manuscripts.

---

## [Referee Report · Reviewer #3 (Public review)]

Summary:

Mitochondrial injury activates eiF2α kinases-PERK, GCN2, HRI and PKR-which collectively regulate the Integrated Stress Response (ISR) to preserve mitochondrial function and integrity. Previous work has demonstrated that stress-induced and pharmacologic stress-independent ISR activation promotes adaptive mitochondrial elongation via the PERK and GCN2 kinases, respectively. Here, the authors demonstrate that pharmacologic ISR inducers of HRI and GCN2 enhance mitochondrial elongation and suppress mitochondrial fragmentation in two disease models, illustrating the therapeutic potential of pharmacologic ISR activators. Specifically, the authors first used an innovative ISR translational reporter to screen for nucleoside mimetic compounds that induce ISR signaling, and identified two compounds, 0357 and 3610, that preferentially activate HRI. Using a mitochondrial-targeted GFP MEF cell line, the authors next determined that these compounds (as well as the GCN2 activator, halofuginone) enhance mitochondrial elongation in an ISR-dependent manner. Moreover, pretreatment of MEFs with these ISR kinase activators suppressed pathological mitochondrial fragmentation caused by a calcium ionophore. Finally, pharmacologic HRI and GCN2 activation was found to preserve mitochondrial morphology in human fibroblasts expressing a pathologic variant in MFN2, a defect that leads to mitochondrial fragmentation and is a cause of Charcot Marie Tooth Type 2A disease.

Strengths:

This well-written manuscript has several notable strengths, including the demonstration of the potential therapeutic benefit of ISR modulation. New chemical entities with which to further interrogate this stress response pathway are also reported. In addition, the authors used an elegant screen to isolate compounds that selectively activate the ISR and identify which of the four kinases was responsible for activation. Special attention was also paid to a thorough evaluation of the effect of their compounds on other stress response pathways (i.e. the UPR, and heat and oxidative stress responses), thereby minimizing the potential for off-target effects. The implementation of automated image analysis rather than manual scoring to quantify mitochondrial elongation is not only practical but also adds to the scientific rigor, as does the complementary use of both the calcium ionophore and MFN2 models to enhance confidence and the broad therapeutic potential for pharmacology ISR manipulation.

Weaknesses:

The only minor concerns are with regard to effects on cell health and the timing of pharmacological administration.

Comments on revisions:

In this revised manuscript the authors demonstrate that pharmacological activation of the eiF2α kinases, HRI and GCN2, induce adaptive mitochondrial elongation and suppress mitochondrial fragmentation in two disease models, illustrating the translational potential of pharmacological ISR modulation.

In revising their manuscript the authors adequately addressed the concerns. In response to comments about the potential toxicity of their compounds, 0357 and 3610, the authors furnish evidence that neither compound significantly reduced viability of HEK293 cells (Figure S1G). Understandably, the authors focused the present work on the acute effects of their compounds. Several other attributes are noteworthy: First, that injury attributable to chronic ISR activation in cell culture may ultimately be circumvented by altering the in vivo pharmacodynamic and pharmacodynamic properties of the compounds, thereby preserving the translation potential for these (and related) compounds. Second, the authors also reasonably explain that the rapidity of ionomycin-induced injury, necessitating that the inducers are administered prior to treatment. Their assessment of the effects of the compounds on mitochondrial fragmentation in MFN2 mutant fibroblasts-in combination with the preserved viability of HEK293 cells-is sufficient to demonstrate the practical pharmacological potential for these (or related) agents.

---

## [Author Response]

The following is the authors’ response to the original reviews.

**Public Reviews:**

**Reviewer #1 (Public review):**
Summary:This manuscript (Baron, Oviedo et al., 2024) builds on a previous study from the Wiseman lab (Perea, Baron et al., 2023) and describes the identification of novel nucleoside mimetics that activate the HRI branch of the ISR and drive mitochondrial elongation. The authors develop an image processing and analysis pipeline to quantify the effects of these compounds on mitochondrial networks and show that these HRI activators mitigate ionomycin-driven mitochondrial fragmentation. They then show that these compounds rescue mitochondrial morphology defects in patient-derived MFN2 mutant cell lines.Strengths:The identification of new ISR modulators opens new avenues for biological discovery surrounding the interplay between mitochondrial form/function and the ISR, a topic that is of broad interest. It also reinforces the possibility that such compounds might represent new potential therapeutics for certain mitochondrial disorders. The development of a quantitative image analysis pipeline is valuable and has the potential to extract the subtle effects of various treatments on mitochondrial morphology.

We thank the reviewer for the positive feedback on our manuscript. We address all of the reviewer’s valuable concerns in the revised submission, as highlighted below.

Weaknesses:I have three main concerns.First, support for the selectivity of compounds 0357 and 3610 acting downstream of HRI comes from using knockdown ISR kinase cell lines and measuring the fluorescence of ATF4-mApple (Figure 1G and 1H). However, the selectivity of these compounds acting through HRI is not shown for mitochondrial morphology. Is mitochondrial elongation blocked in HRI knockdown cells treated with the compounds? While the ISRIB treatment does block mitochondrial elongation, ISRIB acts downstream of all ISR kinases and doesn't necessarily define selectivity for the HRI branch of the ISR. Additionally, are the effects of these compounds on ATF4 production and mitochondrial elongation blocked in a non-phosphorylatable eIF2alpha mutant?

We thank the reviewer for highlighting this point. As indicated by the reviewer, we show that compounddependent increases in mitochondrial elongation are blocked by co-treatment with ISRIB, indicating that this effect can be attributed to ISR activation. We prefer the use of this highly selective pharmacologic approach to block ISR activation, as opposed to the MEF^A/A^ cells, as the use of pharmacologic approaches provide more temporal control over ISR inhibition and can prevent the type of chronic disruption to mitochondria associated with these types of genetic perturbations. However, the reviewer is correct that ISRIB blocks downstream of all ISR kinases, meaning that we cannot explicitly demonstrate that 0357 and 3610 induce mitochondrial elongation downstream of HRI-dependent ISR activation using this tool. Thus, to address this point, we have clarified the discussion of these results to make it clear that our results show that our compounds induce mitochondrial elongation downstream of the ISR, omitting the direct implications of HRI in this phenotype.

This point of selectivity/specificity of the compounds gets at a semantic stumbling block I encountered in the text where it was often stated "stress-independent activation" of ISR kinases. Nucleoside mimetics are likely a very biologically active class of molecules and are likely driving some level of cell stress independent of a classical ISR, UPR, heat-shock response, or oxidative stress response.

A major challenge in defining stress-independent activation of stress-responsive signaling pathways is the fact that the activation of these pathways is often used as a primary marker of cellular stress. While this can be overcome by transcriptome-wide profiling (e.g., RNAseq), the reviewer is correct that our focused profiling of select stress-responsive signaling pathways is insufficient to claim the stress-independent activation of the ISR by our prioritized compounds. To address this, we removed this terminology from the revised submission.

Second, it is difficult for me to interpret the data for the quantification of mitochondrial morphology. In the legend for Figure 2, it is stated that "The number of individual measurements for each condition are shown above." Are the individual measurements the number of total cells quantified? If not, how many total cells were analyzed? If the individual measurements are distinct mitochondrial structures that could be quantified why are the n's for each parameter (bounding box, ellipsoid principal axis, and sphericity) so different? Does this mean that for some mitochondria certain parameters were not included in the analysis? For me, it seems more intuitive that each mitochondrial unit should have all three parameters associated with it, but if this isn't the case it needs to be more carefully described why.

The number of individual measurements refers to the number of 3D segmentations generated using the “surfaces’ module in Imaris. As the reviewer noted, we expect each surface segmentation to represent a single “mitochondrial unit.” We have now further clarified this in the figure legend.

Regarding differences in sample size for each group, we used an outlier test (i.e., ROUT outlier test in PRISM 10) to remove apparent outliers in our data. Often, these outliers result from errors in the automatic quantification that inaccurately merge two mitochondria into one large segmentation. This explains the discrepancy in the number of measurements made for each experimental group. We have made this point more clear in the Materials and Methods section of the revised manuscript.

Third, the impact of these compounds on the physiological function of mitochondria in the MFN2.D414V mutants needs to be measured. Sharma et al., 2021 showed a clear deficit in mitochondrial OCR in MFN2.D414V cells which, if rescued by these compounds, would strengthen the argument that pharmacological ISR kinase activation is a strategy for targeting the functional consequences of the dysregulation of mitochondrial form.

In this manuscript, we demonstrate that pharmacologic activation of the ISR using 0357 and 3610 rescue mitochondrial morphology in patient fibroblasts expressing the disease-associated MFN2^D414V^ mutant. The reviewer is correct that there are other mitochondrial phenotypes linked to the expression of this mutant. We are currently pursuing this question with more potent ISR activating compounds developed in our laboratory identified using the HTS screening platform described in this manuscript. However, this work, which builds on the studies described herein, uses other ISR activating compounds, which we feel would be best described in subsequent manuscripts that can fully define the activity of these new compounds.

**Reviewer #2 (Public review):**
Summary.Mitochondrial dysfunction is associated with a wide spectrum of genetic and age-related diseases. Healthy mitochondria form a dynamic reticular network and constantly fuse, divide, and move. In contrast, dysfunctional mitochondria have altered dynamic properties resulting in fragmentation of the network and more static mitochondria. It has recently been reported that different types of mitochondrial stress or dysfunction activate kinases that control the integrated stress response, including HRI, PERK, and GCN2. Kinase activity results in decreased global translation and increased transcription of stress response genes via ATF4, including genes that encode mitochondrial protein chaperones and proteases (HSP70 and LON). In addition, the ISR kinases regulate other mitochondrial functions including mitochondrial morphology, phospholipid composition, inner membrane organization, and respiratory chain activity. Increased mitochondrial connectivity may be a protective mechanism that could be initiated by pharmacological activation of ISR kinases, as was recently demonstrated for GCN2.A small molecule screening platform was used to identify nucleoside mimetic compounds that activate HRI. These compounds promote mitochondrial elongation and protect against acute mitochondrial fragmentation induced by a calcium ionophore. Mitochondrial connectivity is also increased in patient cells with a dominant mutation in MFN2 by treatment with the compounds.Strengths:(1) The screen leverages a well-characterized reporter of the ISR: translation of ATF4-FLuc is activated in response to ER stress or mitochondrial stress. Nucleoside mimetic compounds were screened for activation of the reporter, which resulted in the identification of nine hits. The two most efficacious dose-response tests were chosen for further analysis (0357 and 3610). The authors clearly state that the compounds have low potency. These compounds were specific to the ISR and did not activate the unfolded protein response or the heat shock response. Kinases activated in the ISR were systematically depleted by CRISPRi revealing that the compounds activate HRI.(2) The status of the mitochondrial network was assessed with an Imaris analysis pipeline and attributes such as length, sphericity, and ellipsoid principal axis length were quantified. The characteristics of the mitochondrial network in cells treated with the compounds were consistent with increased connectivity. Rigorous controls were included. These changes were attenuated with pharmacological inhibition of the ISR.(3) Treatment of cells with the calcium ionophore results in rapid mitochondrial fragmentation. This was diminished by pre-treatment with 0357 or 3610 and control treatment with thapsigargin and halofuginone(4) Pathogenic mutations in MFN2 result in the neurodegenerative disease Charcot-Marie-Tooth Syndrome Type 2A (CMT2A). Patient cells that express Mfn2-D414V possess fragmented mitochondrial networks and treatment with 0357 or 3610 increased mitochondrial connectivity in these cells.

We appreciate the reviewer’s positive response to these aspects of our manuscript. We address the reviewer’s valuable comments in the revised submission as highlighted below.

Weaknesses:The weakness is the limited analysis of cellular changes following treatment with the compounds.(1) Unclear how 0357 or 3610 alter other aspects of cellular physiology. While this would be satisfying to know, it may be that the authors determined that broad, unbiased experiments such as RNAseq or proteomic analysis are not justified due to the limited translational potential of these specific compounds.

The reviewer is correct. The low potency of 0357 and 3610 limit the translational potential for these compounds. However, building on the work described herein, we recently identified more potent HRI activating compounds with higher translational potential. Using RNAseq profiling, we found that these compounds show transcriptomewide selectivity for the ISR and can promote adaptive remodeling of mitochondrial morphology and function in cellular models of multiple other diseases. These compounds will be further described in subsequent studies that expand on the efforts outlined here demonstrating the potential for pharmacologic HRI activators to promote adaptive mitochondrial remodeling.

(2) There are many changes in Mfn2-D414V patient cells including reduced respiratory capacity, reduced mtDNA copy number, and fewer mitochondrial-ER contact sites. These experiments are relatively narrow in scope and quantifying more than mitochondrial structure would reveal if the compounds improve mitochondrial function, as is predicted by their model.

In this manuscript, we demonstrate that pharmacologic activation of the ISR using 0357 and 3610 rescue mitochondrial morphology in patient fibroblasts expressing the disease-associated MFN2^D414V^ mutant. The reviewer is correct that there are other mitochondrial phenotypes linked to the expression of this mutant. We are currently pursuing this question with more potent ISR activating compounds developed in our laboratory using the HTS screening platform described in this manuscript. However, this work, which builds on the studies described herein, uses other ISR activating compounds, which we feel would be best described in subsequent manuscripts that can fully define the activity of these new compounds.

**Reviewer #3 (Public review):**
Summary:Mitochondrial injury activates eiF2α kinases - PERK, GCN2, HRI, and PKR - which collectively regulate the Integrated Stress Response (ISR) to preserve mitochondrial function and integrity. Previous work has demonstrated that stress-induced and pharmacologic stress-independent ISR activation promotes adaptive mitochondrial elongation via the PERK and GCN2 kinases, respectively. Here, the authors demonstrate that pharmacologic ISR inducers of HRI and GCN2 enhance mitochondrial elongation and suppress mitochondrial fragmentation in two disease models, illustrating the therapeutic potential of pharmacologic ISR activators. Specifically, the authors first used an innovative ISR translational reporter to screen for nucleoside mimetic compounds that induce ISR signaling and identified two compounds, 0357 and 3610, that preferentially activate HRI. Using a mitochondrial-targeted GFP MEF cell line, the authors next determined that these compounds (as well as the GCN2 activator, halofuginone) enhance mitochondrial elongation in an ISR-dependent manner. Moreover, pretreatment of MEFs with these ISR kinase activators suppressed pathological mitochondrial fragmentation caused by a calcium ionophore. Finally, pharmacologic HRI and GCN2 activation were found to preserve mitochondrial morphology in human fibroblasts expressing a pathologic variant in MFN2, a defect that leads to mitochondrial fragmentation and is a cause of Charcot Marie Tooth Type 2A disease.Strengths:This well-written manuscript has several notable strengths, including the demonstration of the potential therapeutic benefit of ISR modulation. New chemical entities with which to further interrogate this stress response pathway are also reported. In addition, the authors used an elegant screen to isolate compounds that selectively activate the ISR and identify which of the four kinases was responsible for activation. Special attention was also paid to a thorough evaluation of the effect of their compounds on other stress response pathways (i.e. the UPR, and heat and oxidative stress responses), thereby minimizing the potential for off-target effects. The implementation of automated image analysis rather than manual scoring to quantify mitochondrial elongation is not only practical but also adds to the scientific rigor, as does the complementary use of both the calcium ionophore and MFN2 models to enhance confidence and the broad therapeutic potential for pharmacology ISR manipulation.

We thank the reviewer for their positive response to our manuscript. We address the reviewer’s remaining concerns as outlined below.

Weaknesses:The only minor concerns are with regard to effects on cell health and the timing of pharmacological administration.

The two compounds described in this manuscript were found to not induce any overt toxicity over a 24 h period in cell culture models. In the revised manuscript, we show data showing that treatment with increasing doses of either 0357 or 3610 do not significantly reduce cellular viability in HEK293 cells (Fig. S1G).

With regards to treatments, we include all of the relevant information for the timing and dosage of compound treatment in the revised manuscript.

**Recommendations for the authors:**

**Reviewer #1 (Recommendations for Authors)**
(1) Figure S1 "B. ATF4-Gluc activity" -> Fluc, The number of replicates is not consistently stated for each experiment. p-values are not given for D and F.

We have changed the legend for Fig. S1B to ATF4-FLuc. We show individual replicates for all experiments for all panels described in this figure, except panels C and G, in the revised Figure S1. We explicitly state the number of replicates in panel C and G in the accompanying figure legend. We have repeated the qPCR described in panels C,F and statistics are included in the revised manuscript.

(2) Figure 2 - no p-values for BtdCPU.

Yes. We found that BtdCPU-dependent increases in mitochondrial fragmentation (described in Fig. 2A-D) were not significant when analyzing *all* the data included in these figures by Brown-Forsythe and Welch ANOVA test. However, the DMSO and BtdCPU conditions were significantly different when directly compared using a Welch’s t-test (p<0.005). Since the statistics in this manuscript are being analyzed by ANOVA, we decided not to include a significance marker for BtdCPU, as it was not observed in this more stringent test and is not the main focus of this manuscript.

(3) Figure S4 (Supplement to Figure 5) -> Supplement to Figure 4.

We have corrected this error in the revised manuscript.

(4) Error in references - duplicated 24 and 46, duplicated 10 and 11.

This is now corrected in the revised submission.

**Reviewer #2 (Recommendations for the authors):**
I would love to see an assessment of mitochondrial function and mtDNA in the D414 cells following treatment.

As indicated above, we are continuing to probe the impact of more potent HRI activating compounds in patientderived cell models expressing disease-relevant *MFN2* mutants. Initial experiments suggest that this approach can mitigate additional pathologies beyond deficient elongation in these cells, although we are continuing to pursue these results with our improved HRI activating compounds. We are excited by these results, but feel that they are best suited for a follow-up manuscript describing these new HRI activators.

**Reviewer #3 (Recommendations for the authors):**
The only suggestion to broaden the manuscript's impact might be to perform a basic assessment of the impact of pharmaceutical ISR activation on cell viability. Though mitochondrial elongation is often considered a surrogate for mitochondrial health, whether mitochondrial elongation improves cell viability (or not) would be informative. Similarly, the authors did not address the time-dependent effects of the ISR modulators, choosing to focus on the acute rather more chronic outcomes. Finally, does simultaneous (rather than pre-) treatment with an activator and the ionomycin produce similar effects on mitochondrial morphology, especially since therapeutics are typically administered post-injury?

We now include cell viability experiments showing that the two HRI activators discussed in this manuscript, 0357 and 3610, do not significantly reduce viability in HEK293 cells. This work is included in the revised manuscript (see Fig. S1G).

With respect to acute vs chronic outcomes of ISR activation. As highlighted by the reviewer, we primarily focus this work on defining the impact of acute ISR treatment on mitochondrial morphology. As discussed above, we now show that our prioritized ISR activating compounds 0357 and 3610 do not significantly impact cellular viability over a 24 h timecourse. However, as suggested by the reviewer, additional studies on the potential implications of chronic pharmacologic ISR activation on mitochondrial biology remains to be further explored.

We are continuing to address this in subsequent studies using more potent ISR kinase activating compounds established in our lab. However, we would like to highlight that detrimental phenotypes linked to chronic ISR kinase activation in cell culture does not preclude the translational potential for this approach, as in vivo PK/PD of these compounds can be controlled to prevent complications arising from chronic pathway activity. We previously demonstrated the potential for controlling compound activity through its PK/PD in our establishment of highly selective activators of other stress-responsive signaling pathways such as the IRE1/XBP1s arm of the UPR (e.g., Madhavan et al (2022) *Nat Comm*).

We appreciate the reviewer’s comments regarding the timing of compound treatment in them ionomycin experiment. Ionomycin works extremely quick to induce fragmentation (minutes), which would be prior to activation of the ISR induced by these compounds (hours). Thus, co-treatment would lead to fragmentation. It is an interesting question to ask if co-treatment with ISR activators could rescue this fragmentation as the pathway is activated, but we did not explicitly address this question in this manuscript. However, we do show that pharmacologic GCN2 or HRI activators can rescue mitochondrial morphology in patient fibroblasts expressing a *MFN2* mutant, where mitochondria are fragmented, indicating that our approach can restore mitochondrial morphology in this context. We feel that these results, in combination with others described in our manuscript, demonstrate the potential for this approach to mitigate pathologic mitochondrial fragmentation associated with different conditions.